# Modeling One-Dimensional Touch Pointing with Nominal Target Width

Leave Authors Anonymous

## ABSTRACT

Finger-Fitts law [6] is a variant of Fitts' law which accounts for the finger ambiguity in touch pointing. It involves the effective target width $W_e$ (i.e., $\sqrt{2\pi e}\sigma$) in modeling touch pointing. We hypothesize that the nominal target width ($W$) can be used in lieu of $W_e$ in Finger-Fitts law. Such a model, called Finger-Fitts-$W$ model, complements the original Finger-Fitts law because it suits the situation where the distribution of endpoints is unknown, such as answering the following question without running a study: "what would be the target selection time if the target size increases from 2 to 3 cm?". Although the Finger-Fitts-$W$ model has been implied, it is understudied. In this short paper, we compare using nominal width ($W$) vs. effective width ($W_e$) in one-dimensional touch modeling. The results showed that the Finger-Fitts-$W$ model improves the model fitness over the conventional Fitts' law and has a slight improvement over the original Finger-Fitts law. Our key takeaway is that Finger-Fitts-$W$ is a valid model for predicting touch pointing movement time. It complements the original Finger-Fitts law as it can predict movement time of touch pointing even if the distribution of endpoints is unknown.

**Index Terms:** Human-centered computing—Human computer interaction (HCI)—Interaction techniques—Pointing; Human-centered computing—Human computer interaction (HCI)—HCI theory, concepts and models; Human-centered computing—Human computer interaction (HCI)—Empirical studies in HCI

## 1 INTRODUCTION

Among a number of finger-touch based interaction, pointing has been a dominant input modality on mobile devices such as smartphones and tablets. Due to its prevalence, modeling touch pointing is crucial in designing touch interfaces. Fitts' law [13, 22] (Equation 1), which relates the pointing movement time ($MT$) to the relative precision of the tasks ($\frac{A}{W}$), is the most widely known pointing model. However, despite its success in modeling pointing actions with mouse or stylus, Fitts' law does not address the ambiguity caused by finger touch, which is the widely recognized "fat finger" problem. Hence, it cannot accurately model touch-based pointing.

$$MT = a + b\log_2(\frac{A}{W} + 1). \qquad (1)$$

Finger-Fitts law (a.k.a FFitts law, Equation 2) [6] is a refinement of Fitts' law for modeling touch pointing:

$$MT = a + b\log_2\left(\frac{A}{\sqrt{2\pi e(\sigma^2 - \sigma_a^2)}} + 1\right)$$
$$= a + b\log_2\left(\frac{A}{\sqrt{W_e^2 - 2\pi e\sigma_a^2}} + 1\right). \qquad (2)$$

Previous research [6,34] has shown that Finger-Fitts law (Equation 2) can more accurately model finger-touch pointing than Fitts' law, and has been used for modeling typing speed on soft keyboard [4], for

developing a keyboard decoding algorithm [5], and for modeling other touch interaction such as crossing [21].

The Finger-Fitts law (Equation 2) uses the effective width $W_e$ for modeling, which is calculated from the observed touch points variance ($W_e = \sqrt{2\pi e}\sigma$). Drawing an analogy from Fitts' law research that both effective width ($W_e$) and nominal width $W$ (i.e., the width defined by the geometry of the target) are commonly used to model pointing, we hypothesize that using the nominal target width $W$ in lieu of the $W_e$ in Finger-Fitts law with a small adjustment is also a valid touch pointing model. We call it Finger-Fitts-$W$ model (Equation 3):

$$MT = a + b\log_2\left(\frac{A}{\sqrt{W^2 - c^2}} + 1\right), \qquad (3)$$

where $a$, $b$, and $c$ are empirically determined parameters. Because the Finger-Fitts-$W$ model avoids using the observed touch point variance ($\sigma^2$), it supports predicting the movement time $MT$ without actually carrying out the studies to obtain the variance of touch point distribution ($\sigma^2$). It allows interface designers to ask "what if" questions such as "what would be the target selection time if I increase the target size from 2 cm to 3 cm?". In contrast, the original Finger-Fitts law (referred to as Finger-Fitts-$W_e$ model hereafter) requires to observe the variance of touch point distribution with the new target size to make prediction.

The Finger-Fitts-$W$ model (Equation 3) is also an extension of the recently proposed 2D Finger-Fitts law [19], which uses nominal target width and height in the model. Given the promising performance of 2D Finger-Fitts law, it is likely that using the nominal task parameter $W$ in lieu of the observed touch points variance ($W_e = \sqrt{2\pi e}\sigma$) is also valid for modeling one-dimensional touch pointing. Although such a model has been implied, it is not explicitly expressed nor studied, especially in the context of one-dimensional touch pointing.

To fill this knowledge gap, in this short paper, we explicitly express the Finger-Fitts-$W$ model (Equation 3), and present a study comparing the effective width ($W_e$) vs. nominal-width ($W$) in Finger-Fitts law for modeling one-dimenional touch pointing. Our investigation showed that Finger-Fitts-$W$ model performed the best among tested models including Fitts' law and Finger-Fitts-$W_e$ model, showing that Finger-Fitts-$W$ model is valid for modeling touch pointing. Although it is only a small adjustment over the original Finger-Fitts law [6], it is rather necessary and advances our understanding of touch pointing. It also generalizes the nominal parameter based two-dimensional Finger-Fitts law model [19] to one-dimensional touch pointing.

## 2 RELATED WORK

We review related work on (1) using Fitts' law and its variants to model pointing, and (2) modeling finger touch pointing with Finger-Fitts law.

### 2.1 Modeling 1D pointing

As one of the best known theoretical foundations of HCI, Fitts' law (Equation 1) [13, 22] has served as a cornerstone for interface and input device evaluation [9, 22], interface optimization [20], and interaction behavior modeling [11].

The beauty of the original Fitts' law lies in its simplicity. It is a pure task model of human pointing performance, in which all of the

model's independent variables are *a priori* task parameters $A$ and $W$. For a given graphical object's distance and size, for example, designers can predict or estimate the average time it takes a user to complete a pointing task at it.

One challenge of applying Fitts' law is that a user might or might not comply with the task precision defined by $A/W$ when performing the tasks, causing over- or under-utilization of target width [35]. This is partly because a user may adopt different speed-accuracy trade-off policies [3, 4, 15, 16, 23, 25, 33]. The way researchers have addressed the varied degree of task compliance is to bend Fitts' law away from a pure task model towards a behavioral one by changing an independent variable in the model from a task parameter $W$ (target width) to "effective width", an *a posterior* quantity depending on user's behavior. First proposed by Crossman [12] and explored further [22, 26, 32], the effective width adjustment method has shown stronger model fit if the observed error rates deviate from 4%. It replaces the nominal target width $W$ with the so-called effective width $W_e$ (i.e., $\sqrt{2\pi e}\sigma$), as shown in Equation 4.

$$MT = a + b \cdot \log_2\left(\frac{A}{\sqrt{2\pi e}\sigma} + 1\right) \qquad (4)$$

$$= a + b \cdot \log_2\left(\frac{A}{W_e} + 1\right), \qquad (5)$$

Controlled studies [35] showed that using $W_e$ could partially but not fully account for the subjective layer of speed-accuracy trade-off. Involving the posterior variable $\sigma$ complicates Fitts' law as a predictive tool for design. Later in the next section we explain in detail that because the Fitts' law with effective width adjustment (Equations 4 and 5) is the basis of Finger-Fitts law [6]), the limitation of involving a posterior variable also limits the predictive power of Finger-Fitts law.

Another line of Fitts' law research closely related to the current work is about modeling small-sized target acquisition tasks. Previous researchers [32] have proposed using $W - c$ instead of $W_e = \sqrt{2\pi e}\sigma$ to adjust the target width in Fitts' law, where $c$ was an experimentally determined constant attributed to hand tremor. The modified version gave a good fit for both pencil-based [32] and mouse-based [10] pointing tasks. Our research later shows that $c$-constant model could serve as a simplification of the refined Finger-Fitts model, with reduced model fitness.

## 2.2 Modeling finger touch pointing

As finger touch has become the dominant input modality in mobile computing, a sizable amount of research has been carried out to understand and model the uncertainty in touch interaction. On a capacitive touchscreen, a touch point is converted from the contact region of the finger. This is an ambiguous and "noisy" procedure, which inevitably introduces errors. Factors such as finger angle [17, 18] and pressure [14] may affect the size and shape of the contact region, unintentionally altering the touch position. The lack of visual feedback on where the finger lands due to occlusion (the "fat finger" problem) further exacerbates the issue [17, 18, 27–29]. As a result, it is hard to precisely control the touch position even with fine motor control ability.

This "fat finger" problem, or the lack of absolute precision in finger touch, presented a challenge to use Fitts' law as a model for finger touch-based pointing, because the only variable in Fitts' law, namely Fitts' index of difficulty, $log_2(A/W + 1)$, is solely determined by the relative movement precision, or the distance to target size ratio.

Bi, Li and Zhai [6–8] identified this challenge, and proposed the Finger Fitts law [6] to address it. They derived their model by separating two sources of end point variance - those due to the absolute imprecision of finger touch (denoted by $\sigma_a{}^2$) and those due to the speed-accuracy trade-off demonstrated in a pointing process

(denoted by $\sigma_r{}^2$). The end point variance caused by the imprecision of finger touch ($\sigma_a{}^2$) is irrelevant to the speed-accuracy trade-off so it should be accounted for. They accounted for it by subtracting $\sigma_a{}^2$ from the observed variance $\sigma^2$, which led to Finger-Fitts law (Equation 2). Following the notation of effective width $W_e = \sqrt{2\pi e}\sigma$ (or $4.133\sigma$) [12, 26, 32], Finger-Fitts law (Equation 2) can be re-expressed as Equation 6:

$$MT = a + b\log_2\left(\frac{A}{\sqrt{W_e{}^2 - 2\pi e\sigma_a{}^2}} + 1\right). \qquad (6)$$

Later research [4, 6, 21, 34] showed that Finger-Fitts law was successful in modeling touch interaction. For example, research [4] showed it was more accurate than the typical Fitts' law in estimating the upper bound of typing speed on a virtual keyboard. Researchers [21] extended the Finger-Fitts law to the crossing action with finger touch, which improved the model fitness ($R^2$) from 0.75 to 0.84 over the original Fitts' law. The recent work [19] extends Finger-Fitts law from 1D to 2D, which shows using nominal target width and height is valid for modeling 2-dimensional touch pointing. Complementary to the previous work [19], this work investigates modeling 1-dimensional target selection with nominal target widths. We also compare effective width vs. nominal width while the previous work [19] did not draw such a comparison.

As alluded to earlier, previous research on Finger-Fitts law is mostly based on using the effective width $W_e$. Next, we describe how we use the nominal width $W$ in Finger-Fitts law (a.k.a the Finger-Fitts-$W$ model), and present a study comparing it with using effective width and the typical Fitts' law.

## 3 USING NOMINAL WIDTH $W$ TO MODEL TOUCH POINTING

To use nominal width $W$ in touch modeling, a straightforward approach is to replace the effective width $W_e$ in Finger-Fitts-$W_e$ mode (Equation 2) with $W$:

$$MT = a + b\log_2\left(\frac{A}{\sqrt{W^2 - 2\pi e\sigma_a^2}} + 1\right) \qquad (7)$$

A potential problem is that it leaves the equation undefined if $W < \sqrt{2\pi e}\sigma_a$. To address this problem, we assume that $\sigma_a^2$, which represents the absolute variance caused by finger touch, is an empirical parameter determined from the data, instead of a pre-defined constant:

$$MT = a + b\log_2\left(\frac{A}{\sqrt{W^2 - c^2}} + 1\right), \qquad (8)$$

where $a$, $b$, $c$ are all empirically determined parameters. The implication of this adjustment is that $\sigma_a^2$ may differ across task contexts, and treating it as a free parameter would provide more flexibility in modeling. The drawback is that it introduces an extra free parameter $c$ to the model. In the model evaluation, we take the number of free parameters into consideration and control for the overfitting.

Replacing $W_e$ with $W$ also has a physical meaning. The $W_e$ represents the observed variance in the endpoint distribution, which is the actual endpoint variability a user exhibits. In contrast, $W^2$ represents the endpoint variability allowance specified by the task parameter, which is the variability allowance a user is supposed to consume.

## 4 EVALUATION IN 1D POINTING TASKS

We carried out an experiment to investigate whether the Finger-Fitts-$W$ law can accurately predict $MT$, compared with Fitts' law and the original Finger-Fitts law. The study was a reciprocal target acquisition task with finger touch.

### 4.1 Participants and Apparatus

We recruited 14 subjects for an IRB approved study (3 females; aged from 22 - 35). All of them were right-handed and daily smartphone users. A Google Pixel C tablet with 2560x1800 resolution and 308 ppi was used throughout the experiment. Each participant was instructed to perform the tasks on the tablet. They were instructed to select the target with the index finger as fast and accurately as possible. All the subjects were daily smartphone users.

### 4.2 Design and Data Processing

#### 4.2.1 Target Acquisition Task

We designed a within-subject reciprocal target acquisition task for circular targets with various diameters. We chose circular target acquisition mainly because of two reasons. First, this is one of the common tasks used in Fitts' law studies (e.g., [6, 24]). Second, $\sigma_a = 1.5$ is obtained from the circular target acquisitions in [6]. Adopting a similar circular targets experiment setting allows us to investigate not only vertical finger traveling but horizontal movements.

The study included 15 conditions with 5 levels (4, 6, 8, 10, 12 mm) of diameters ($W$) and 3 levels (16, 28, 60 mm) of distance ($A$). It had two different movement directions, which are vertical and horizontal movements. Each condition included 20 touches (19 trials, where the first touches in each condition are considered the starting action) and the condition would show up in random order. We have 14 (participants) $\times$ 15 (conditions) $\times$ 2 (directions) $\times$ 19 (trials) = 7,980 (trials) in total.

At the beginning of each trial, two circular targets were displayed on the touch screen, one in red (a.k.a the start circle) and one in blue (a.k.a the destination circle). The participant was instructed to select the start circle to start the trial. Upon successfully selecting the start circles, the colors of start and destination circles got swapped and the participant was instructed to select the destination circle as fast and accurately as possible. A successful sound would be played if the target was successfully selected. Otherwise, a failure sound was played. The elapsed time between the moment the user successfully selected the start circle and the moment the user subsequently landed down the touch point to select the destination circle was recorded as the movement time of the current trial; the touch point for selecting the destination circle was the location of the endpoint, regardless of whether the touch point was within or outside the target boundary. If the participant succeeded in selecting the destination circle, the colors of two circles were swapped again and the next trial started immediately. If the participant failed in selecting the destination circle, she had to successfully select it again to start the next trial. This setting ensured that in each trial the finger always starts from somewhere within the starting circle, reducing the noise in measuring $A$.

#### 4.2.2 Data processing

We pre-processed the data by removing touch points which fell beyond 3 standard deviations to the target center. In circular acquisition tasks, 50 out of 7,980 touch points (0.63%) were removed as outliers.

### 4.3 Results

#### 4.3.1 $MT$ and error rates across the condition

We observe movement time and the error rates across different target widths and distances (Table 1 and 2).

For movement times, a repeated measure ANOVA test showed that both width $W$ ($F_{4,52} = 175.3$, $p < 0.0001$) and distance $A$ ($F_{2,26} = 320.7$, $p < 0.0001$) had a statistically significant effect. The interaction effect of width and distance was also significant ($F_{8,104} = 2.077$, $p < 0.05$). For error rates, a repeated measure ANOVA test showed that width $W$ had a significant effect ($F_{4,52} = 56.19$, $p < .0001$), but not distance $A$ ($F_{2,26} = 1.443$, $p = 0.255$).

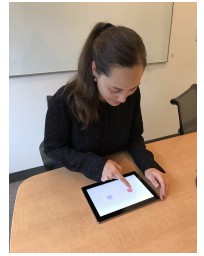
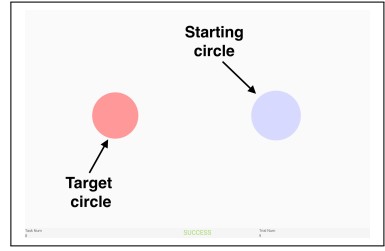

(a) Demonstration    (b) Targets (Circular)

Figure 1: (a) A participant was doing the task. (b) A screenshot of the task.

The interaction effect of width and distance was not significant ($F_{8,104} = 1.965$, $p = 0.058$).

The results also showed that participants were more error-prone with smaller targets, especially with diameters under 6 mm. A repeated measure ANOVA test showed that the size of the targets (targets which are 4 and 6 mm are considered small targets) had a significant effect ($F_{1,13} = 80.21$, $p < 0.0001$). This results concurred with conclusion from other research [6, 8].

| Diameters (mm) | MT Mean (SD) | Error rate |
|---|---|---|
| 4 | 0.50 (0.13) | 24.9% |
| 6 | 0.37 (0.13) | 10.9% |
| 8 | 0.31 (0.11) | 6.4% |
| 10 | 0.28 (0.10) | 2.8% |
| 12 | 0.25 (0.09) | 1.1% |

Table 1: Movement time and error rates over different target widths

| Distances (mm) | MT Mean (SD) | Error rate |
|---|---|---|
| 16 | 0.26 (0.11) | 8.2% |
| 28 | 0.31 (0.12) | 9.3% |
| 60 | 0.45 (0.13) | 10.0% |

Table 2: Movement time and error rates over different distances

#### 4.3.2 Regression for $MT$ vs. $ID$

Figure 2 shows the regression results of $MT$ vs. $ID$. As shown, the Finger-Fitts-$W$ law has the highest $R^2$ value (0.986) among all the test models, indicating its high model fitness. The results also showed that Finger-Fitts-$W_e$ model was better than the typical Fitts' law - $W$ and the Fitts' law - $W_e$, consistent with findings from previous work [6].

#### 4.3.3 RMSE of $MT$ Prediction

To increase the external validity of the evaluation, we also examine the Root Mean Square Error (RMSE) of $MT$ prediction with cross validation. We conduct leave-one-($A$, $W$)-out cross validation and obtain the RMSE for Finger-Fitts-$W$, Fitts' law - $W_e$ and Fitts' law - $W$. The results (Table 2) are 0.015 for Finger-Fitts-$W$ model, 0.021 for Finger-Fitts-$W_e$ model, 0.033 for Fitts' law - $W$, and 0.064 for Fitts' law - $W_e$. It showed Finger-Fitts-$W$ outperformed all the other three models.

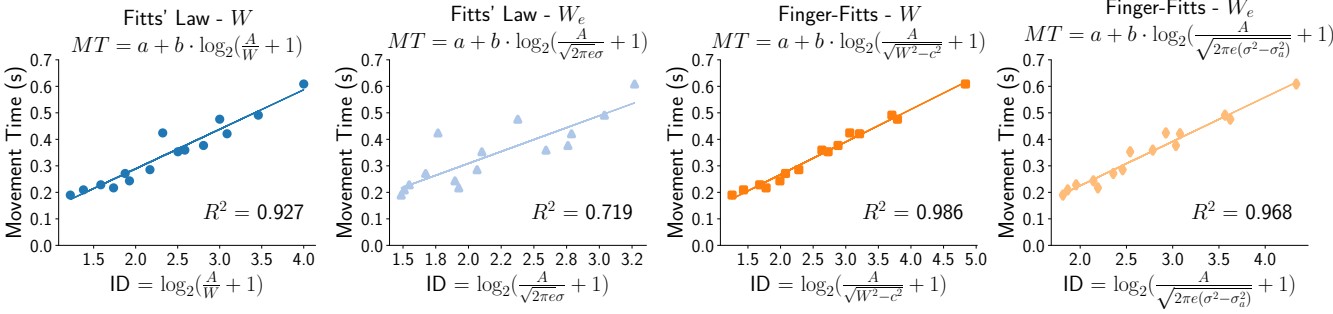

Figure 2: *MT vs. ID* regressions for Fitts' law, Fitts' law with effective width, Finger-Fitts-*W*, and Finger-Fitts-*W*$_e$ models. As shown, Finger-Fitts-*W* model shows the best model fitness.

| | | | $R^2$ | RMSE | AIC | WAIC | Parameters |
|---|---|---|---|---|---|---|---|
| ● | Fitts' Law - *W* | Eq. (1) | 0.927 | 0.033 | -94.00 | -86.87 | $a = -0.009, b = 0.149$ |
| ▲ | Fitts' Law - *W*$_e$ | Eq. (4) | 0.719 | 0.064 | -73.82 | -69.21 | $a = -0.051, b = 0.180$ |
| ■ | Finger-Fitts Law - *W* | Eq. (8) | 0.986 | 0.015 | -118.81 | -109.46 | $a = 0.021, b = 0.123, c^2 = 11.260$ |
| ◆ | Finger-Fitts Law - *W*$_e$ | Eq. (2) | 0.968 | 0.021 | -106.37 | -100.35 | $a = -0.109, b = 0.167, \sigma_a = 1.5$ |

Table 3: The parameters, $R^2$, *RMSE* of leave-one-$(A, W)$-out cross validation, and Information Criteria AIC and WAIC of the models. For AIC and WAIC, the smaller the values, the more accurate the model prediction.

### 4.3.4 Information Criteria

Information criteria [1, 2, 30, 31] have been widely used to compare the quality of models because they take into account the complexity of the model (i.e., the number of parameters). Commonly used information criteria include *AIC*, and *WAIC*, both of which penalize the complexity of a model. In general, the smaller the information criterion, the better the model is. We have calculated multiple information criteria including *AIC* and *WAIC* (Table 3). As shown, the Finger-Fitts-*W* law outperforms the Fitts' law - *W* and Finger-Fitts-*W*$_e$ law in these metrics.

## 4.4 Discussion

### 4.4.1 The validity of the Finger-Fitts-*W* model.

Our study showed that the Finger-Fitts-*W* law had the highest prediction accuracy among all the test models across a number of metrics, including information criteria that take into account the number of model parameters. The $R^2$ value, a commonly used measure for pointing model fitness, is 0.986, higher than both Fitts' law - *W* ($R^2 = 0.927$) and Finger-Fitts-*W*$_e$ ($R^2 = 0.968$). Its RMSE, a cross-validation metric, is also the lowest among all the test models. To take into account the number of model parameters, we examined the information criteria. The AIC and WAIC showed that the Finger-Fitts-*W* law improved prediction accuracy over Fitts' and Finger-Fitts law. It reduced AIC from -94.00 to -118.81, and WAIC from -86.87 to -109.46. These two metrics added penalty to adding extra parameters in the Finger-Fitts-*W* law, adding evidence to the strength of the Finger-Fitts-*W* model.

### 4.4.2 The "*w* − *c*" model serves as a simplification of Finger-Fitts-*W* model.

We notice that the Finger-Fitts-*W* law resembles the model proposed in 1968 by Welford [32] in which $W - c$ was used in lieu of $W$ to account for the hand tremor when acquiring small-sized targets with pencil (i.e., $MT = a + b \log_2(\frac{A}{W-c} + 1)$). We investigated whether this $W - c$ model could serve as a simplification of the Finger-Fitts-*W* model based on the data collected in our experiment. Our analysis

shows that the $W - c$ model has slightly weaker prediction performance than the Finger-Fitts-*W* law, with $R^2 = 0.984$ and RMSE = 0.016. Its performance is still better than the Fitts' law - *W* and Finger-Fitts-*W*$_e$ law. Therefore, it can serve as a simplification of the Finger-Fitts-*W* law for touch modeling.

## 5 CONCLUSIONS

Our main conclusion is that the one-dimensional Finger-Fitts-*W* model (Equation 9), a variant of the Finger-Fitts law [6] can model the movement time in touch pointing quite well:

$$MT = a + b \log_2 \left( \frac{A}{\sqrt{W^2 - c^2}} + 1 \right). \quad (9)$$

It complements the original Finger-Fitts law [6] which predicts movement time with the variance of observed touch point distribution. Because it uses nominal parameters only for prediction, the Finger-Fitts-*W* model can answer "what if" questions without obtaining the variance of touch point distribution. Our evaluation shows the Finger-Fitts-*W* model outperforms Fitts' law (which also uses nominal parameters only for prediction) in model fitness, measured by $R^2$ values, cross-validation *RMSE*, and information criteria. Overall, our investigation shows that the Finger-Fitts-*W* model is a valid model for modeling one-dimensional touch pointing.

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
