# OpenReview forum: "Modeling One-Dimensional Touch Pointing with Nominal Target Width"
_graphicsinterface.org/Graphics_Interface/2021/Conference — Submitted to GI 2021_

### Official Review · AnonReviewer3 · 2021-01-11
**A short, well-presented modification to FFitts Law**

**Rating:** 7
**Confidence:** 2

**Review:**

This paper presents a modified version of the established FFitts law equation, which replaces the effective width component with width and a third coefficient, c. The authors present a study to validate their model, and compare the complexity of their model with the original FFitts equation, as well as two popular versions of Fitts' Law.

I realize that the benefit of not specifically calculating effective width. The coefficient c is calculated a priori and can model hypothetical designs. However, are you not just hiding effective width measures in the "empirically determined" c? Furthermore, what is the procedure to measure and calculate c?

The paper is well-written and well-presented throughout. I appreciate the authors' inclusion of their model parameters and study procedure. Measuring the information criteria of the models is a nice touch, but I suggest writing out the acronyms AIC and WAIC when introduced in 4.3.4.

I would like to see other additional studies to validate Finger-Fitts-W before accepting it, but I realize that publishing this short paper is an ideal way for researchers to investigate it further.

---

### Official Review · AnonReviewer2 · 2021-01-12
**Small contribution is fine, but several issues with context and analysis**

**Rating:** 5
**Confidence:** 5

**Review:**

This 4-page paper reports on a single experiment to test a revised version of FFitts Law [6] that uses nominal width instead of effective width, and an empirically determined third parameter instead of a constant value representing variance from absolute touch accuracy.  The results suggest the refined model is better, primarily based on lower AIC/WAIC and lower RMSE. The model itself is not new, and the work overall is very close to a recent UIST 2020 paper [19], but this is ok.  I have no problem accepting a paper to GI that makes a small contribution with a single experiment to provide some additional and refined empirical evidence for previous work.

I really wanted to argue for accept, but I found several things that concern me. First, in spite of the title, this isn’t a 1D task in terms of target size, it’s a 1D movement for a 2D target. Second, the closely related [19] is not described in enough detail and the model tested in the submission is really exactly the same as the “smaller-than” model tested in [19].  Third, the main metrics used to compare the FFitts-W and FFitts-We and form the main contribution have apparent issues: AIC might be calculated as though FFitts-WE has 3 params instead of 2, and RMSE comparison uses no statistical test and the very small different size isn’t acknowledged. Finally, the performance of FFitts-We is largely dependent on the sigma_a parameter, but the submission uses one calculated on an old phone in [6].  Perhaps all of these could be clarified and fixed with revisions, but maybe these are errors that undermine the validity and size of this already small contribution.  Unfortunately, as the paper is submitted I can’t argue for accept.

Although this paper is all about a 1D task, it is not according to the standard definition used in pointing. A true 1D task is when the target is constrained in only one direction, the other target dimension is infinite. The constrained dimension is most typically an “amplitude” constraint, meaning it is aligned with direction of movement (see Accot and Zhai’s 2002 “More than dotting the i's” paper for a good explanation of 1D pointing with different direction constraints). This is exactly how a 1D task is defined in the original FFitts paper [6]. This submission seems to conflate a 1D task to be only be about the direction of travel, but since the target is a circle, the task is 2D regardless of the movement directions tested. In fact, the paper used the “2D” version of the sigma_a parameter instead of the “1D” one in [6]. This seems to further acknowledge the tested task isn’t really 1D.

The very similar and closely related paper [19] should be described in more detail and more accurately. The current description is essentially two sentences in the intro and one sentence in the related work. In both places, the model and work of [19] are characterized simply as extending FFitts to 2D and using nominal width and height.  But there are important details about [19] not covered, such that it tested two variations of FFitts models for 2D targets, “Euclidean” and “smaller-of”.  The results showed the smaller-of model was best which is quite relevant to this submission, because the “1D” task (where W and H are the same) means that the smaller-of model with nominal width trivially reduces to the “FFitts-W” model (eq 8) in this submission.  In other words, the models are the same.

The submission also doesn’t properly acknowledge that the “c” parameter introduced in [32] (to avoid using sigma_a) was also tested in [19] with touch. Section 3 of seems to be suggesting that using c for touch in the submission is something new for touch, but it isn’t.

Comparing models using AIC is great, much better that simply comparing R^2 values as was done in older papers. The main benefit of an AIC-based comparison is a penalty for adding more model parameters. This means if two models are similar in terms of fit, but one has fewer parameters, it will have smaller AIC. This raises a concern because Table 3 seems to suggest that FFitts-We has three parameters by also considering sigma_a. But sigma_a is a constant value for the regression (the submission is clear that sigma_a value is always 1.5 taken from [6]). If AIC and WAIC are calculated as though FFitts-We has 3 parameters, then those measures are incorrect, and given how close they are, one of the main results of the paper may be wrong.

It isn’t stated, but I believe RMSE units must be in seconds. Therefore, the differences between the RMSE for the main FFitts-W and FFitts-We models are quite small: about 6ms or 1.2% of a typical 0.5s pointing task. Given RMSE values are calculated from several runs, the stddev or 95%CI of these average RMSE values should be reported, and a stats test to verify a claim like “FFitts-W outperformed all other models” for RMSE.  Some discussion about the practical relevance of such a small effect of 6ms difference should also be included.

I wonder if using the sigma_a value from [6] is a good choice. That was measured on old phone technology, a Nexus with 480x800 resolution at 254PPI. The touch sensor hardware on a Pixel C tablet is likely much better, and it has a 21% higher PPI. This means a sigma_a on the Pixel C might be more accurate. [6] states that the sigma_a values they suggest should be verified in future work. A better approach would have been to include the simple and fast calibration task used in [6] to determine the best sigma_a for the Pixel C.

This is a small point, but the ANOVA analysis in 4.3.1 is not very useful considering the goal of the paper. It only shows obvious effects of W and H on time and error. In addition, the analysis isn’t complete without reporting posthoc tests.  I’m guessing this was included because it’s always reported, or perhaps to demonstrate that the study task was sound. Perhaps a sentence to explain the purpose of this analysis would help to guide the reader, or even remove it and explain to the reader why.

---

### Official Review · AnonReviewer1 · 2021-01-13
**Good, small contribution - but some questions about the information criteria**

**Rating:** 5
**Confidence:** 3

**Review:**

This work discusses a modification on the Finger-Fitts (FFitts) model, and a study which demonstrates that it slightly outperforms the previous model (despite an additional regression-fitted parameter). The new model seems fairly sensible, but not particularly novel - for instance, it is very close to an old model by Welford, as cited by the authors of the present study. The main contribution here seems to be the evidence showing that this model is more suitable for predicting the movement times for finger touch events.

I would have appreciated a quick explanation as to how $c$ was fitted to the data - whether it was determined directly through some mathematical procedure (as $a$ and $b$ often are), or whether some optimization strategy was used to determine its optimal value. In the former case, providing a formula for $c$ would be useful for future practictioners.

I appreciate the authors' use of the AIC and WAIC to assess the impact of their additional parameter c. This helps addresses a major potential criticism of the work - otherwise, the addition of an extra parameter could significantly improve the fit without actually being meaningful or useful simply due to increased fitting power. I have a few small comments about the way they've employed the information criteria. First, I believe the authors should also report the BIC (Bayesian Information Criteria); both the AIC and BIC are commonly used to evaluate model suitability and have different statistical properties. Second, I question whether $\sigma_a$ should qualify as a parameter in FFitts-We, given that it is set as the constant 1.5 and is not adjusted through model fit (unlike the $c$ parameter of FFitts-W). If $\sigma_a$ is fitted, as opposed to simply fixed by a previous study, the comparison between FFitts-W and FFitts-We would become more valid. If the IC values are recalculated with two parameters for FFitts-We, I believe the AIC/WAIC values will come closer to that of FFitts-W, and the BIC values may even show FFitts-We slightly ahead.

Although the work itself has a fairly small contribution, the short page length (4) merits consideration for inclusion, especially as the contribution is notable and focused. Given the importance of the information criteria analysis to this contribution, I would definitely like to see a revision to the paper to include the necessary corrections and additional analysis.

---

### Meta-Review · Area_Chair1 · 2021-01-15

**Recommendation:** Reject
**Confidence:** 4

**Metareview:**

The paper was reviewed by three reviewers, one of whom self-rated their confidence as 5/5 (expert). The two reviewers with the highest confidence scores also rated the paper slightly below the acceptance threshold.

R3 was the most positive, providing a short review which praises the paper, calling it well-written and well-presented. R1 is likewise positive, noting that the "short page length merits consideration for inclusion", but only after certain revisions are made (thus, not arguing for acceptance of the current version).

R2 gave a very long and comprehensive review covering several drawbacks of the current work. Both R1 and R2 note that the described model is very similar to existing models (R2 actually notes that it is identical to a model from [19]), and that there are some issues with the AIC/WAIC computations that might compromise the conclusion. R2 also notes that the paper depends on a $\sigma_a$ value which is significantly out-of-date.

Of these issues, I feel that the AIC/WAIC issue is potentially the most serious, as it compromises the authors' position that their technique outperforms the existing FFitts-We model. In short, the computation of AIC/WAIC as if FFitts-We had three parameters, rather than the two it functionally has, biases the computations in favor of their 3-parameter FFitts-W model. Given that the paper's (small) contribution partially hinges on this result, it seems like it would need to be fixed or rectified to recommend acceptance.

I am therefore recommending that the paper be rejected from this round of submissions. However, based on the feedback from all reviewers, it seems like a few changes to the paper could push it towards acceptance, and thus I recommend that the authors consider revising and resubmitting to the second round.

---

### Decision · Program_Chairs · 2021-01-16

Reject